# Regulation of Monocyte Perilipin-2 Expression in Acute and Chronic Coronary Syndromes: Pathogenetic Implications

**DOI:** 10.3390/ijms26199550

**Published:** 2025-09-30

**Authors:** Francesco Canonico, Renzo Laborante, Chiara Pidone, Ramona Vinci, Mattia Galli, Eugenia Pisano, Alice Bonanni, Marianna Di Sario, Anna Severino, Lucia Lisi, Daniela Pedicino, Giovanna Liuzzo, Massimiliano Ruscica, Filippo Crea, Giuseppe Patti, Domenico D’Amario

**Affiliations:** 1Department of Thoracic and Cardiovascular Diseases, Azienda Ospedaliero-Universitaria Maggiore Della Carità, 28100 Novara, Italy; 2Department of Cardiovascular and Pneumological Sciences, Catholic University of Sacred Heart, 00168 Rome, Italy; 3Department of Medical-Surgical Sciences and Biotechnologies, Sapienza University of Rome, 04100 Latina, Italy; 4GVM Care & Research, Maria Cecilia Hospital, 48033 Cotignola, Italy; 5Department of Cardiovascular Sciences, Fondazione Policlinico Agostino Gemelli IRCCS, 00168 Rome, Italy; 6Department of Pharmacology, Catholic University of Sacred Heart, 00168 Rome, Italy; 7Department of Pharmacological and Biomolecular Sciences, University of Milan, 20133 Milan, Italy; 8Department of Cardio-Thoracic-Vascular Diseases, Foundation IRCCS Cà Granda Ospedale Maggiore Policlinico, 20152 Milan, Italy; 9Center of Excellence of Cardiovascular Sciences, Ospedale Isola Tiberina—Gemelli Isola, 00186 Rome, Italy; 10Department of Translational Medicine, Università del Piemonte Orientale, 28100 Novara, Italy

**Keywords:** PLIN2, STEMI, CCS, proteasome activity, oxidative stress, plaque instability

## Abstract

PLIN2 is involved in the lipid metabolism of macrophages resident in atherosclerotic plaques, and its upregulation leads to lipid droplets (LDs) accumulation. LDs enlargement results in the macrophage transformation into foam cells, a key step for the onset of atherosclerosis. In the present study, we investigated the role of PLIN2 and its regulation mechanisms in atherosclerosis and plaque instability in patients with a diagnosis of ST-elevation myocardial infarction (STEMI) and chronic coronary syndrome (CCS). We enrolled STEMI (*n* = 122) and CCS patients (*n* = 45). Peripheral blood mononuclear cells were isolated from whole blood samples. The PLIN2 protein level was analyzed in CD14+ monocytes by flow cytometry. Lipidomic panel and proteasome activity were evaluated. PLIN2 protein expression was significantly correlated with the age of CAD patients. We found no significant difference in monocyte lipid content between the two patient groups. The PLIN2 increased in STEMI as compared to CCS patients (*p* < 0.001). The proteasome activity being higher in STEMI as compared to CCS patients (*p* < 0.001), significant inverse correlations were evident between PLIN2 levels and proteasome activity in the CCS groups (*p* = 0.02). PLIN2 expression was higher in STEMI as compared to CCS patients, suggesting an involvement in plaque instability. Despite the proteasome activity being higher in STEMI patients, probably due to the elevated inflammatory burden, PLIN2 could escape proteasome degradation in a more efficient manner in STEMI as compared to CCS patients.

## 1. Introduction

Atherothrombotic cardiovascular diseases account for over 4 million deaths annually in Europe [1], underscoring the critical need for further research into their causes and prevention. Atherosclerotic lesions in large arteries are characterized by foam cells, which derive from macrophages that uptake large amounts of low-density lipoprotein (LDL) and oxidized-LDL (ox-LDL) through phagocytosis and scavenger receptor-mediated pathways, leading to lipid droplet (LD) formation [2]. These LDs serve as the primary site of intracellular cholesterol storage, consisting of a hydrophobic core of neutral lipids, mainly cholesteryl esters, surrounded by a phospholipid monolayer with associated proteins [3]. In macrophages and foam cells, Perilipin-2 (PLIN2), also known as adipose differentiation-related protein (ADRP), is the key LD-coating protein. It promotes cholesterol retention and prevents efflux, thus playing a pivotal role in atherogenesis.

PLIN2’s involvement in atheroma development has been well-documented, both in cultured macrophages and the murine models of atherosclerosis [4]. The overexpression of PLIN2 in macrophages or THP-1 cells enhances LD accumulation and increases pro-inflammatory cytokine production, while its downregulation decreases lipid content and prevents LD formation [5,6]. Furthermore, PLIN2 gene inactivation reduces foam cell lipid loading and attenuates atherosclerosis in animal models [7,8]. PLIN2 also serves as a marker of lipid loading and is implicated in neo-atherogenesis following drug-eluting stent implantation, as well as in microcirculation obstruction in ST-segment elevation myocardial infarction (STEMI) patients [9,10].

The regulation of PLIN2 occurs at the post-translational level via proteolytic degradation, involving both the ubiquitin/proteasome pathway and chaperone-mediated autophagy (CMA) [11]. The ubiquitin–proteasome system, particularly the 26S proteasome, regulates PLIN2 in foam cells and liver cells [12,13,14]. In contrast, CMA facilitates the chaperone-dependent sorting of cytosolic proteins to lysosomes for degradation, with lysosome-associated membrane protein 2A (LAMP2A) serving as the rate-limiting factor in this process [15].

However, the precise differences in PLIN2 expression and function in acute coronary syndrome (ACS) versus chronic coronary syndrome (CCS) remain unclear. This study aims to address this gap by investigating the differential regulation of PLIN2 and its role in atherosclerosis and plaque instability, particularly in patients diagnosed with STEMI and CCS.

## 2. Results

### 2.1. PLIN2 Protein Expression in Chronic and Acute Coronary Syndromes

PLIN2 protein levels resulted higher in STEMI patients compared to CCS patients (0.94 ± 0.21 vs. 0.77 ± 0.13 median fluorescence intensity (MFI); *p* < 0.0001) (Figure 1).

### 2.2. Lipidomic Panel

No significant differences in monocyte lipidomes were observed between STEMI and CCS patients (STEMI *n* = 10; CCS *n* = 10; and *p =* n.s.) (Figure 2). Appendix A reports the LM IDs, names, systematic names, and formulas of the analyzed lipids (Appendix A).

### 2.3. Proteasome Activity in STEMI and CCS Patients

LAMP2A was not different in the two groups [STEMI vs. CCS 1.960 (1.490–2.880) vs. 2.025 (1.410–2.423); *p =* n.s.] and was not correlated with PLIN2 protein expression (*p =* n.s.) (Figure 3A,B). Proteasome activity was higher in STEMI as compared to CCS patients (STEMI vs. CCS 3.788 ± 1.078 vs. 2.372 ± 1.327; *p* < 0.001), and a significant inverse correlation was evident between PLIN2 levels and proteasome activity in the CCS group (*p =* 0.018), while it was not significant in the STEMI group (Figure 3C,D).

### 2.4. STEMI Patients Had Higher Levels of Oxidative Stress Compared to CCS

STEMI patients showed an increase in serum ROS/RNS values compared to CCS patients (STEMI vs. CCS 68.61 ± 10.23 vs. 57.26 ± 10.54; *p =* 0.0067) (Figure 4).

### 2.5. Age-Related PLIN2 Expression in CAD Patients

PLIN2 expression was significantly correlated with age in STEMI (*r*^2^ = 0.09; *p =* 0.002) and CCS patients (*r*^2^ = 0.24; *p =* 0.014) (Figure 5). When performing a univariate analysis of variance, with variable PLIN2 of STEMI and CCS patients in relation to age, with linearly independent pairwise comparisons between the estimated marginal means, the difference in the two correlations was found to be significant (*p =* 0.003). No correlation was found with other clinical variables such as the body mass index (BMI), total cholesterol (TC), low-density lipoprotein cholesterol (LDL-C); high-density lipoprotein cholesterol (HDL-C); triglycerides (TG); glycemia; and C-Reactive Protein (CRP) (Appendix A). The current findings suggest a potential role for PLIN2 in contributing to age-related atherosclerosis and its progression [16,17]. No significant correlations were observed between the PLIN2 level or proteasomal activity and other potentially influencing clinical factors, such as lipid-lowering therapy, inflammatory status, and comorbidities.

## 3. Discussion

### 3.1. PLIN2 and Coronary Instability

Our data provide novel evidence that STEMI patients exhibit higher monocyte PLIN2 protein expression compared to CCS patients, suggesting an emerging potential role for PLIN2 in coronary instability, likely driven by its pro-inflammatory effects. This finding aligns with prior studies linking PLIN2 to local inflammation in atherosclerotic plaques [18]. Specifically, PLIN2 overexpression has been associated with an increased production of pro-inflammatory cytokines, such as tumor necrosis factor-α, monocyte chemoattractant protein-1, and interleukin-6, in macrophages activated by acetylated LDL [6]. Additionally, Norman et al. demonstrated that the in vitro inhibition of monocyte PLIN2 reduced the pro-inflammatory and pro-atherogenic gene expression induced by triglyceride-rich lipoprotein lipolysis products [19].

These observations suggest that heightened local plaque inflammation due to PLIN2 overexpression could promote thrombus formation and increase the thrombus burden following plaque destabilization, thereby impairing plaque healing mechanisms [20,21]. Future studies should investigate the relationships of PLIN2 values with peri- and post-procedural procedures, especially considering PLIN2 levels in relation to intracoronary imaging and the dynamic assessment of ischemic hemorrhagic risk, particularly in patients with greater plaque instability [22]. Furthermore, increased local inflammation may exacerbate oxidative stress in the infarct-related coronary vessel, which could reduce the availability of nitric oxide and disrupt vasodilation mechanisms in the coronary microcirculation, ultimately heightening the risk of microvascular obstruction [22]. These findings underscore the potential of PLIN2 as a potential key modulator in coronary instability and a promising therapeutic target to mitigate thrombotic complications and microvascular dysfunctions in STEMI patients.

### 3.2. PLIN2-Related Molecular Mechanisms

The increased levels of PLIN2 in STEMI patients, despite increased proteasome activity, suggests the presence of molecular mechanisms that potentially render the protein resistant to this type of degradation. The main mechanisms on which research should focus its efforts are post-translational modifications, for example, the potential phosphorylation and acetylation of PLIN2, making it less recognizable and more stable, respectively; cellular compartmentalization, for example, the PLIN2 fractions associated with LDs and free cytosolic fractions; and finally, interactions with other regulatory factors that could make PLIN2 more stable. This increased stability could allow PLIN2 to evade its primary degradation mechanism, highlighting the need for further investigation of the underlying regulatory mechanisms.

### 3.3. Clinical Implications

Aging is a prevailing risk factor for the formation of clinically significant atherosclerotic lesions, and it is associated with a gradual decrease in plaque stability; however, age-related mechanisms underlying plaque instability still remain unclear [16,17]. A significant correlation of PLIN2 expression with age in coronary artery disease (CAD) patients was found, suggesting that it might represent a control mechanism disturbed by age. A decline of the proteostasis capacity during aging leads to the dysfunction of specific cell types and tissues, rendering the organism susceptible to a range of chronic diseases.

Abundant evidence links aberrant PLIN2 expression to the severity of various metabolic and age-related diseases, including fatty liver, insulin resistance, and atherosclerosis—conditions unified by lipid accumulation and chronic inflammation [8,23,24,25]. Notably, PLIN2 has recently been identified as a key player in the pathogenesis of nonalcoholic steatohepatitis (NASH), with studies showing that hepatic PLIN2 deficiency mitigates lipid accumulation, inflammation, and fibrosis in diet-induced mouse models of hepatic steatosis. These findings position PLIN2 as a potential bridge connecting lipid accumulation and inflammation [26,27], offering a promising target for therapeutic intervention.

An efficient turnover of PLIN2, regulated by CMA and the proteasome, is critical for proper lipid metabolism. In the liver, impaired CMA leads to increased PLIN2 levels and a corresponding decline in the LD breakdown by cytosolic adipose triglyceride lipase (ATGL). However, analysis of LAMP2A, the rate-limiting factor of CMA, revealed no significant differences between STEMI and CCS patients, suggesting that CMA is not involved in PLIN2 dysregulation in this clinical context. While CMA-mediated PLIN2 regulation has been well-documented in the liver, its role in monocytes remains debated [11].

Our findings, however, underscore the proteasome as a primary regulator of PLIN2 in monocytes, as evidenced by the inverse relationship between PLIN2 and proteasome levels in CCS patients. This insight is clinically relevant, as it points to potential therapeutic avenues targeting the proteasome to modulate PLIN2 levels and mitigate lipid-induced inflammation in atherosclerosis and other metabolic diseases. Future studies are needed to investigate the causes and molecular mechanisms underlying degradation resistance, such as post-translational modifications, cellular compartmentalization, and interactions with other regulatory factors. Additionally, recent studies have suggested that oxidative stress can induce PLIN2 expression in hepatocytes, further linking this protein to cellular stress responses. HepG2 cells treated with hydrogen peroxide showed increased PLIN2 mRNA and protein expression, which could offer valuable insights into the potential role of oxidative stress in regulating PLIN2 in cardiovascular and metabolic diseases [28].

Our data revealed that cells from STEMI patients exhibited higher intracellular oxidative stress compared to CCS patients, suggesting a dysregulated mechanism contributing to the elevated PLIN2 levels. The lipidomic analysis of monocytes from the enrolled patients did not show significant differences between the STEMI and CCS groups. However, despite the lack of significant lipidomic variation, the marked difference in PLIN2 protein levels between the two groups suggests that the proteasomal escape of PLIN2 may not be directly related to the lipid composition of the monocytes. Future studies should compare the lipid profiles of monocytes from healthy controls to assess whether lipid composition is linked to PLIN2 protein levels.

In conclusion, this study proposes that PLIN2 may escape proteasomal degradation pathways, potentially due to a more stable state in peripheral blood mononuclear cells (PBMCs) of STEMI patients compared to CCS patients. This increased stability could allow PLIN2 to evade its primary degradation mechanism, highlighting the need for further investigation into the underlying regulatory mechanisms.

### 3.4. Study Limitations

Our study has several limitations, first, the mechanisms underlying PLIN2 escape from proteasomal degradation processes are not clarified, therefore future studies in this direction are necessary. In this study, the evaluation of proteasome activity was evaluated in PBMCs; future studies will have to evaluate it selectively on CD14+ monocytes to have a more specific result for the monocyte population only [29,30,31,32]. Second, the post-transcription and post-translation mechanisms of PLIN2, cellular compartmentalization, interactions with other regulatory factors, and the functional analyses of CMA have not been investigated; further studies are needed to clarify this aspect for a better understanding of the degradation mechanisms. Third, the unbalanced sample size between the two study populations represents a further limitation of the study. We also acknowledge that the exclusion of NSTEMI patients could be a further limitation. However, this decision was intentional, as the NSTE-ACS population is highly heterogeneous in terms of clinical presentation, mechanisms underlying plaque instability, and time of presentation. Therefore, the inclusion of these patients could have introduced additional variability and potential bias in the interpretation of our results. In contrast, STEMI patients represent a more homogeneous group in terms of acute presentation and pathophysiology, thus allowing for clearer interpretation of the data. Fourth, we acknowledge that the lipidomic analysis was performed on a relatively small subset of patients (10 per group), which indeed represents a limitation and may restrict the generalizability and robustness of these specific findings. This choice was primarily driven by the complexity, cost, and resource requirements associated with detailed lipidomic profiling. Nevertheless, we believe that even with the limited sample size, the analysis provides valuable exploratory insights and supports the overall biological plausibility of our findings. Fifth, peri- and post-procedural strategies such as imaging-guided PCI and optimized antithrombotic regimens represent important clinical value in patients undergoing PCI with atrial fibrillation or in patients with increased plaque instability potentially related to increased PLIN2 expression [33]. We acknowledge the lack of integration of intravascular imaging techniques such as OCT or IVUS as a limitation, which would greatly improve the translational relevance of our findings by allowing direct correlations between PLIN2 levels and plaque morphology or vulnerability characteristics. We also recognize this approach as a promising avenue for future studies in which the combination of molecular markers, such as PLIN2, with high-resolution intravascular imaging could provide deeper mechanistic insights and potentially improve risk stratification in patients with coronary artery disease. Finally, we acknowledge that, given the observational and cross-sectional nature of our study, it is not possible to establish causality between PLIN2 expression and coronary instability or plaque events. In future studies, a longitudinal assessment of PLIN2 levels in relation to treatment (e.g., statins, anti-inflammatory therapy) could clarify whether PLIN2 represents a modifiable risk marker and could be useful for improved risk stratification.

These results represent preliminary findings that should be interpreted as hypothesis generating. Further mechanistic studies are needed to clarify the underlying mechanisms, integrating the methodology with intravascular imaging for the assessment of plaque stability.

## 4. Materials and Methods

### 4.1. Study Population

In the present study, we prospectively enrolled 167 patients with the following conditions: STEMI (*n* = 122) and CCS (*n* = 45). The STEMI patient is characterized by the complete occlusion of a coronary artery and by a greater severity, in the context of ACS. In order to obtain a more homogeneous and uniform study group, and for greater reliability of the data, we enrolled, for one of the two arms, only STEMI patients. The diagnosis was based on European Society of Cardiology guidelines for STEMI [34] and CCS [35]. The clinical and demographic characteristics and the Laboratory tests of the enrolled patients are reported in Table 1. The Laboratory tests present in Table 1 were performed by the Central Laboratory of the Fondazione Policlinico Universitario Agostino Gemelli IRCCS as per normal clinical practice, such as Glucose level, TC, LDL-C, HDL-C, TG, Hemoglobin, Platelets, white blood cells (WBC), Lymphocytes, Neutrophil, and monocytes.

The diagnosis of STEMI was based on the presence of typical chest pain lasting more than 30 minutes and which was unresolved by administration of intravenous isosorbide dinitrate (2–4 mg), ST-segment elevation ≥ 0.2 mV in at least two contiguous leads, or new onset of left bundle branch block in the initial electrocardiogram and elevation of cardiac troponin values with at least one value above the 99th percentile upper reference limit. Exclusion criteria were as follows: patients with systemic diseases such as autoimmune disorders, acute and chronic infections, renal failure greater than stage III, according to KDIGO classification, liver diseases, neoplasms, and blood diseases; surgical interventions or major traumas in the last 3 months; and patients undergoing rescue percutaneous coronary intervention (PCI) or with evidence of stent thrombosis or late presentation.

CCS patients, with angiographically confirmed CAD, no previous ACS, no overt ischemic episode in the preceding 48 hours, and with atherosclerosis as an inclusion criterion were enrolled. Exclusion criteria were age < 18 years; evidence of inflammatory or infectious diseases, neoplasms, and immunological or hematological disorders; and allergic disorders.

Clinical features were carefully recorded, including demographic data, classical cardiovascular risk factors (smoke, diabetes mellitus, hypertension, dyslipidemia, and familial history), report of previous ACS, past coronary revascularization procedures, angiographic findings, left ventricular ejection fraction (LVEF), and medical treatment at the time of enrolment. This study was approved by the relevant Ethics Committee of our Institution. Informed written consent was obtained from all participants included in the study.

### 4.2. Monocytes Isolation

Following enrolment, patients in both study arms had peripheral venous blood sampling performed. PBMCs were isolated from EDTA whole blood samples through standard gradient centrifugation over Ficoll-Hypaque (GE Healthcare Bio-Sciences, Piscataway, NJ, USA). Monocytes were purified by CD14 purification Kit (Miltenyi, Bologna, Italy) using direct magnetic labeling of PBMCs, according to manufacturer’s instructions. Isolated monocytes were washed, centrifuged, and stored at −80 °C for further analysis.

### 4.3. Flow Cytometry Analysis

PBMCs were incubated with fluorochrome-conjugated monoclonal Abs anti-CD14 ECD (Beckman Coulter, Brea, CA, USA). For intracellular analysis, PBMCs were fixed and permeabilized and then incubated with fluorochrome-conjugated mAbs anti-PLIN2 Alexafluor-488 (Abcam, Cambridge, UK) and anti-LAMP2A (Abcam, Cambridge, UK). Unstained cells were used as a negative control. Flow cytometry analysis was conducted with FC 500 (Beckman Coulter, Brea, CA, USA) and CytoFLEX (Beckman Coulter, Brea, CA, USA), and data were analyzed with Kaluza software (Beckman Coulter, Brea, CA, USA).

### 4.4. Lipidomic Panel

The pellets of the previously isolated monocytes were lysed in 20 μl of NaOH, diluted in water, and a protein assay was performed. A first extraction phase was obtained by the use of the solution (MTBE:MEOH 3:1) pre-cooled at −20 °C for 30 min. After incubation at 4 °C for 45 min and shaking at 160 rpm, a cold sonication was carried out for 15 min. Subsequently, a separation solution (H2O:MeOH 3:1) was added, centrifuged, and the upper layer of the solvent was transferred to a new falcon, frozen, and subsequently analyzed with MS scan 2-1000 ES+ (Waters corporation, Milford, MA, USA).

### 4.5. Proteasome Activity

PLIN2 protein levels are regulated by CMA and proteasome. To assess the role of these protein homeostatic mechanisms on PLIN2 expression, LAMP2A expression and proteasome activity were evaluated in monocytes of STEMI and CCS patients. Cell pellets were re-suspended in 150 µl lysis buffer (10 mM Hepes, pH 7.9, 10 mM KCl, 1.5 mM MgCl_2_, and 1 mM DTT) and incubated for 30 min on ice. Cell homogenate was centrifuged at 13,000 rpm for 15 min at 4 °C. Proteasome activity was assayed using AMC-tagged peptide substrate (Suc-LLVY-AMC), which releases free highly fluorescent AMC (Ex/Em 350/440 nm) in the presence of proteolytic activity. Varioskan™ LUX multimode microplate reader (Thermo Scientific™, Waltham, MA, USA) was used for fluorescence analysis.

### 4.6. Oxidative Stress Quantification

Serum samples were isolated from whole blood and used for oxidative stress quantification. DCF ROS/RNS Assay kit was used for measuring the total free radical assessment, according to the manufacturing protocol (Abcam, Cambridge, UK). Varioskan™ LUX multimode microplate reader (Thermo Scientific™, Waltham, MA, USA) was used for fluorescence analysis.

### 4.7. Statistical Analysis

Data were described as mean ± standard deviation or median with interquartile range (IQR), based on their distribution. For comparisons between groups, we used the Mann–Whitney test if the results did not pass the test for a normal distribution, otherwise, we used a parametric t-test; for categorical variables, a χ^2^ test or Fisher’s exact test was used, as appropriate. For lipidomic panel analysis, we used Two-Way ANOVA with Sidak’s multiple comparisons tests. A *p* value < 0.05 was considered statistically significant. Statistical analysis was performed with GraphPad Prism version 8.0.2 (GraphPad Software, San Diego, CA, USA).

## 5. Conclusions

Our study identifies a correlation between the PLIN2 protein expression and age in CAD patients, with higher PLIN2 levels observed in STEMI compared to CCS patients. PLIN2 expression appears to be linked to oxidative stress and metabolism-driven inflammation, both contributing to coronary instability. Although lipidomic analysis of monocytes revealed no significant differences between groups, an inverse correlation between proteasomal degradation and PLIN2 levels was seen in CCS but not in STEMI patients. Elevated oxidative stress and inflammation in STEMI likely facilitate PLIN2’s escape from proteasomal degradation, independent of lipid composition. The potential clinical implications related to PLIN2 regulation may include its potential prognostic and therapeutic use, for example, in the early identification and risk stratification for the onset and progression of CAD or as a potential biomarker for better management of patient drug treatments. Future longitudinal studies will be needed to understand whether PLIN2 is a modifiable risk marker and useful for better risk stratification. Findings suggest PLIN2 plays a significant role in lipid metabolism, and also potentially in coronary instability and patient prognosis, presenting an opportunity for targeting PLIN2 to improve outcomes in atherosclerotic disease.

## Figures and Tables

**Figure 1 ijms-26-09550-f001:**
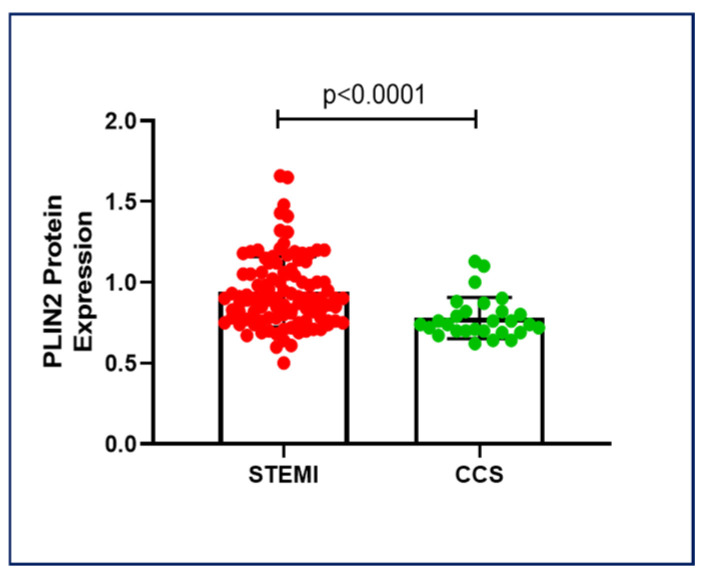
PLIN2 protein expression in STEMI and CCS patients. STEMI patients showed higher PLIN2 protein levels than CCS patients, suggesting potential PLIN2-related key pathways in coronary instability. Abbreviations: perilipin-2 (PLIN2); ST-elevation myocardial infarction (STEMI); and chronic coronary syndrome (CCS).

**Figure 2 ijms-26-09550-f002:**
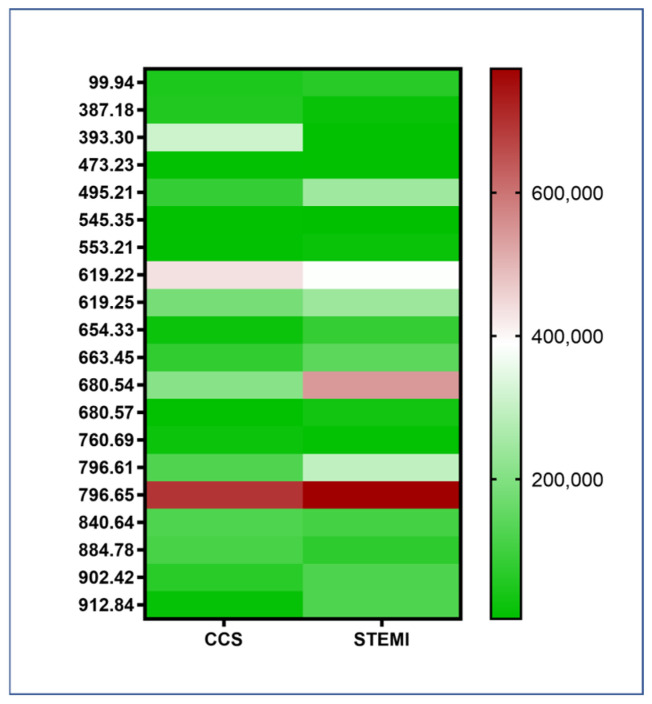
Heatmap of the lipid panel in monocytes of CAD patients. The heatmap with lipid peak area values analyzed shows no significant differences between STEMI and CCS patients and no connection between the type of lipid stored in the monocyte and the amount of PLIN2 protein. Abbreviations: ST-elevation myocardial infarction (STEMI); and chronic coronary syndrome (CCS).

**Figure 3 ijms-26-09550-f003:**
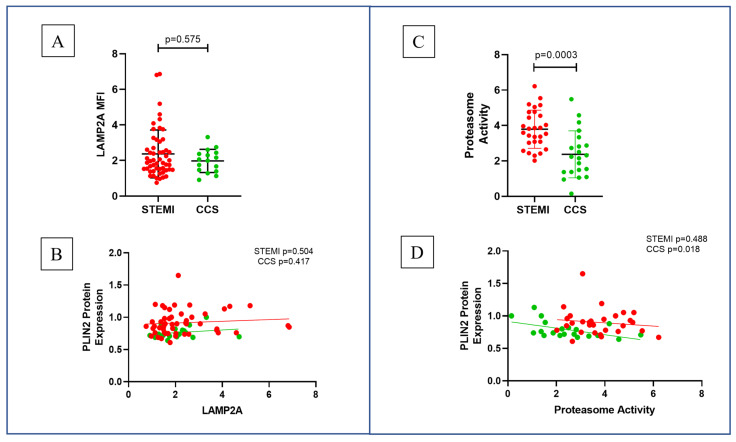
Mechanisms of PLIN2 degradation. (**A**) No difference was found in LAMP2A expression and (**B**) LAMP2a-PLIN2 correlation between the two groups; (**C**) proteasome activity was significantly increased in STEMI patients compared to CCS patients; and (**D**) proteasome activity was correlated with PLIN2 protein expression in CCS patients, but not in STEMI patients. Abbreviations: perilipin-2 (PLIN2); lysosome-associated membrane protein 2A (LAMP2A); median fluorescence intensity (MFI); ST-elevation myocardial infarction (STEMI); and chronic coronary syndrome (CCS).

**Figure 4 ijms-26-09550-f004:**
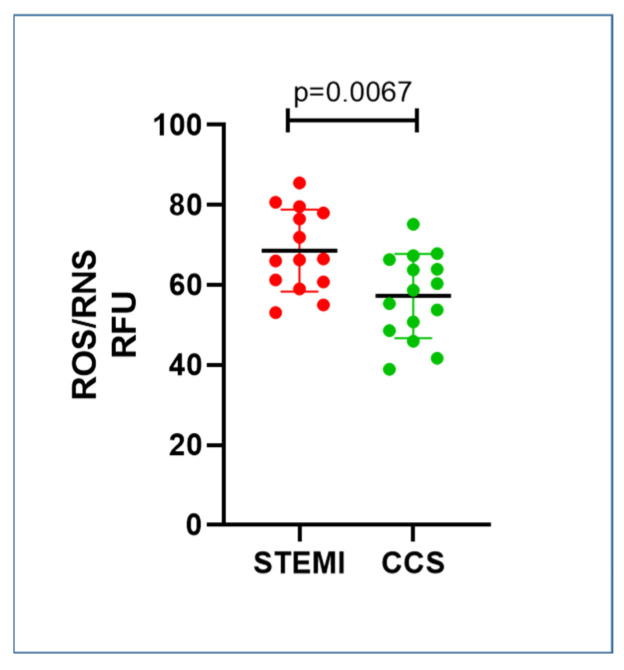
Serum oxidative/nitrosative stress values in STEMI and CCS patients. Scheme STEMI patients displayed a higher oxidative stress intracellular concentration compared to CCS patients, suggesting a further dysregulated mechanism underlying the increased PLIN2 levels. Abbreviations: reactive oxygen species (ROS); reactive nitrogen species (RNS); and relative fluorescence units (RFU); ST-elevation myocardial infarction (STEMI); and chronic coronary syndrome (CCS).

**Figure 5 ijms-26-09550-f005:**
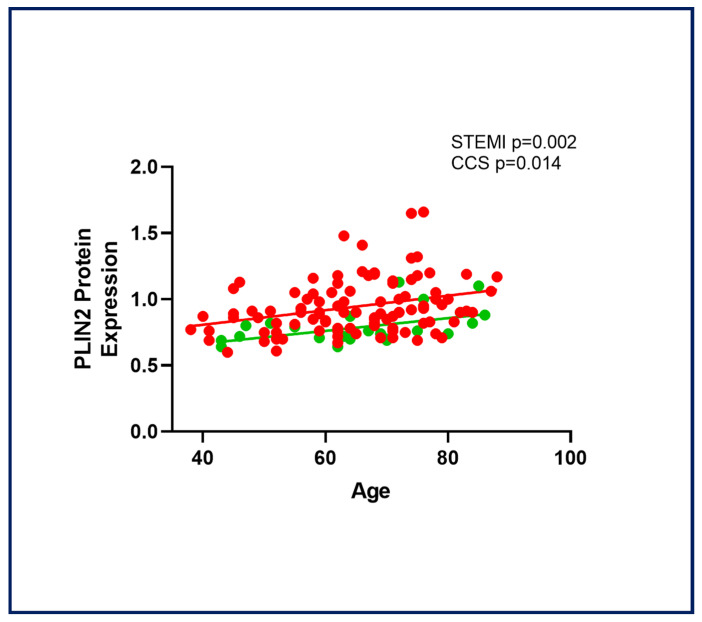
PLIN2 protein expression was correlated with age in CAD patients. In a univariate analysis of variance, the difference in the two age-related PLIN2 correlations in STEMI and CCS patients was significant. Abbreviations: perilipin-2 (PLIN2); ST-elevation myocardial infarction (STEMI); and chronic coronary syndrome (CCS).

**Table 1 ijms-26-09550-t001:** Clinical and demographic characteristics of the enrolled patients. This table shows the clinical characteristics of the enrolled patients. The *p*-values are shown. *p*-values less than 0.05 are considered significant.

	Overall	STEMI	CCS	*p*-Value
Number of Patients	167	122	45	
Sex (M/F)	138/29	100/22	38/7	*p* = 0.8202
Age (Median—IQR)	68 (59–74)	67 (57–74)	69 (61–76)	*p =* 0.2495
Weight (Mean ± SD)	77.4 ± 14.2	78.4 ± 15.1	76.0 ± 11.7	*p =* 0.3559
Height (Mean ± SD)	1.71 ± 0.1	1.71 ± 0.1	1.71 ± 0.1	*p =* 0.8072
BMI (Mean ± SD)	26.4 ± 4.0	26.7 ± 4.25	25.8 ± 3.23	*p =* 0.2397
**Risk factors [*n* (%)]**				
Dyslipidemia	81 (49)	55 (45)	26 (59)	*p =* 0.1588
Hypertension	104 (63)	73 (60)	31 (70)	*p =* 0.2761
Smoker	85 (52)	65 (54)	20 (45)	*p =* 0.3818
Family History of IHD	53 (32)	35 (29)	18 (41)	*p =* 0.1867
Diabetes	39 (24)	26 (21)	13 (30)	*p =* 0.3039
Obesity	18 (11)	14 (12)	4 (9)	*p =* 0.7827
**Pharmacological treatment (at the time of blood sampling) [*n* (%)]**				
Aspirin	55 (35)	27 (24)	28 (62)	*p* < 0.0001 *
Antiplatelet Drug	16 (10)	10 (9)	6 (13)	*p =* 0.5598
Anticoagulants	21 (13)	12 (11)	9 (20)	*p =* 0.1242
Beta-blockers	43 (27)	23 (20)	20 (44)	*p =* 0.0030 *
Diuretics	29 (18)	14 (12)	15 (33)	*p =* 0.0033 *
ACEi	32 (20)	23 (20)	9 (20)	*p* > 0.9999
Sartans	33 (21)	20 (18)	13 (29)	*p =* 0.1324
Statins	56 (35)	37 (32)	19 (42)	*p =* 0.2688
Ca-antagonists	29 (18)	19 (17)	10 (22)	*p =* 0.4955
Insulin	10 (6)	6 (5)	4 (9)	*p =* 0.4721
OHA	29 (18)	20 (18)	9 (20)	*p =* 0.8204
**Laboratory assay (Mean ± SD)**				
Glucose Level (mg/dL)	122 ± 51.7	128 ± 56.2	105 ± 31.3	*p =* 0.0009 *
TC (mg/dL)	166 ± 38.9	169 ± 41.2	160 ± 32.5	*p =* 0.4627
LDL-C (mg/dL)	101 ± 33.1	103 ± 35.0	95 ± 26.0	*p =* 0.2688
HDL-C (mg/dL)	42 ± 10.9	40 ± 10.0	48 ± 11.4	*p =* 0.0001 *
TG (mg/dL)	124 ± 86.1	131 ± 97.0	106 ± 45.9	*p =* 0.2154
Hemoglobin (g/dL)	14 ± 8.44	14 ± 1.90	13 ± 1.87	*p =* 0.1319
Platelets (10^9^/L)	233 ± 70.63	237 ± 74.22	222 ± 59.01	*p =* 0.4214
WBC (10^9^/L)	9.11 ± 3.9	9.84 ± 4.1	7.19 ± 2.6	*p* < 0.0001 *
Lymphocytes (10^9^/L)	1.96 ± 0.97	1.99 ± 1.07	1.88 ± 0.62	*p =* 0.9947
Neutrophil (10^9^/L)	7.10 ± 3.28	7.91 ± 3.23	4.91 ± 2.26	*p* < 0.0001 *
Monocytes (10^9^/L)	0.57 ± 0.28	0.61 ± 0.30	0.46 ± 0.16	*p =* 0.0028 *

Abbreviations: standard deviation (SD); body mass index (BMI); ischemic heart disease (IHD); angiotensin converting enzyme inhibitors (ACEi); oral hypoglycemic agent (OHA); total cholesterol (TC); low-density lipoprotein cholesterol (LDL-C); high-density lipoprotein cholesterol (HDL-C); triglycerides (TG); and white blood cells (WBC). * *p* < 0.05.

## Data Availability

The original contributions presented in this study are included in the article/Appendix A. Further inquiries can be directed to the corresponding author.

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
