# Peer review of "Regulation of Monocyte Perilipin-2 Expression in Acute and Chronic Coronary Syndromes: Pathogenetic Implications"

_ijms, 2025, doi:10.3390/ijms26199550_

Round 1
Reviewer 1 Report
Comments and Suggestions for Authors
This manuscript presents an interesting analysis of the role of PLIN2 in monocyte/macrophage lipid metabolism in patients with acute and chronic coronary artery disease. This topic is undoubtedly of interest to the scientific community studying atherosclerosis and plaque vulnerability. The study stands out for its translational approach, linking molecular and cellular data to a clinically well-characterized cohort of patients with STEMI and CCS.
Clarify the rationale for the correlation between PLIN2 and proteasome activity: the observation of increased PLIN2 expression despite increased proteasome activity in STEMI patients is interesting. It would be useful to further discuss the possible molecular mechanisms underlying resistance to degradation (e.g., post-translational modifications, cellular compartmentalization, or interactions with other regulatory factors).
Examine the lipidomic data: although no significant differences in lipid content were found between the two groups, it would be useful to clarify which lipid classes were analyzed and whether there were non-significant trends consistent with the hypothesis of selective accumulation in STEMI patients.
Control for confounding variables: The correlation found with age in patients with CAD is noteworthy, but it would be useful to know whether other clinical factors (e.g., lipid-lowering therapy, inflammatory status, comorbidities) were considered or excluded as potential confounders in the assessment of PLIN2 and proteasomal activity.
Extended discussion of clinical impact: The results suggest that PLIN2 may be a potential biomarker of plaque instability. It would be interesting to extend the discussion to include possible clinical implications (e.g., prognostic or therapeutic use) and compare your results with existing studies on PLIN2 in cardiovascular settings.
Author Response
This manuscript presents an interesting analysis of the role of PLIN2 in monocyte/macrophage lipid metabolism in patients with acute and chronic coronary artery disease. This topic is undoubtedly of interest to the scientific community studying atherosclerosis and plaque vulnerability.
Thank you for the kind comment.
The study stands out for its translational approach, linking molecular and cellular data to a clinically well-characterized cohort of patients with STEMI and CCS.
Thank you for the kind comment.
Clarify the rationale for the correlation between PLIN2 and proteasome activity: the observation of increased PLIN2 expression despite increased proteasome activity in STEMI patients is interesting. It would be useful to further discuss the possible molecular mechanisms underlying resistance to degradation (e.g., post-translational modifications, cellular compartmentalization, or interactions with other regulatory factors).
We sincerely thank the reviewer for this valuable comment. We acknowledge the importance of additional experimental validation; however, due to resource constraints, we are unable to perform further experiments at this stage.
To address this limitation, we have substantially revised the manuscript as follows:
- New mechanistic discussion – We have added a dedicated paragraph discussing in greater detail the potential underlying mechanisms that may explain the observed findings (see revised manuscript, page 3, lines 133-141). This increased stability could allow PLIN2 to evade its primary degradation mechanism, highlighting the need for further investigation into the underlying regulatory mechanisms."
- Revised figure – A new version of Graphical Abstract has been included, illustrating the possible mechanistic events in a schematic manner.
- Explicit acknowledgement – We have emphasized this limitation both in the Discussion section (page 4, lines 166-168) and in the dedicated Limitations paragraph (page 4, lines 189-192) acknowledging that " We have added the need to investigate these mechanisms in the section on clinical implications.
We believe these revisions improve the clarity and completeness of the manuscript while transparently acknowledging the limitations of the present study.
Examine the lipidomic data: although no significant differences in lipid content were found between the two groups, it would be useful to clarify which lipid classes were analyzed and whether there were non-significant trends consistent with the hypothesis of selective accumulation in STEMI patients.
Lipidomic analysis was carried out using UPLC-MS/MS. The choice was made to set an ESI+ SCAN and check for the presence of all extracted lipids with a m/z transition from 50/1000 in a 15-minute run. With these molecular weights, there are many lipids and therefore many peaks. The final analysis focused more on evaluating the profile rather than on the specific mass/lipid. The data for the studied lipids (LM_ID; Name; Systematic Name and Formula) are reported in the attached "Table S1.".
Control for confounding variables: The correlation found with age in patients with CAD is noteworthy, but it would be useful to know whether other clinical factors (e.g., lipid-lowering therapy, inflammatory status, comorbidities) were considered or excluded as potential confounders in the assessment of PLIN2 and proteasomal activity.
We thank the reviewer for this insightful comment. We carefully assessed the potential influence of additional clinical factors, including lipid-lowering therapy, inflammatory status, and comorbidities. No significant correlations were observed between these variables and either PLIN2 expression or proteasomal activity.
This information has now been added to the revised manuscript (page 3, lines 109-110) to clarify that age was the only factor showing a noteworthy association in our cohort.
However, we acknowledge that the lack of correlation with other variables might in part be related to the relatively small number of patients included in the study, and we have highlighted this limitation in the revised version.
However, I report the p-values of the correlations considering only patients whose proteasomal activity was assessed below:
Statin [STEMI 0.5788 vs CCS 0.3739]
CRP [STEMI 0.2158 vs CCS 0.7997]
Hypertension [STEMI 0.8679 vs CCS 0.0833]
Diabetes [STEMI 0.3632 vs CCS 0.7415]
Smoking [STEMI 0.6446 vs CCS 0.6081]
Dyslipidemia [STEMI 0.0828 vs CCS 0.4713]
Obesity [STEMI 0.9095 vs CCS 0.6016]
Familiarity [STEMI 0.9122 vs CCS 0.5419]
Extended discussion of clinical impact: The results suggest that PLIN2 may be a potential biomarker of plaque instability. It would be interesting to extend the discussion to include possible clinical implications (e.g., prognostic or therapeutic use) and compare your results with existing studies on PLIN2 in cardiovascular settings.
Thank you for this comment. We have added the following section to the "Concluding remarks" paragraph: "Potential clinical implications related to PLIN2 regulation may include its potential prognostic and therapeutic use, for example, in early identification and risk stratification of disease progression."

Reviewer 2 Report
Comments and Suggestions for Authors
In this prospective study, Dr. Francesco Canonico and colleagues aimed to investigate the role of Perilipin-2 (PLIN2) and its regulation mechanisms in atherosclerosis and plaque instability in patients with a diagnosis of ST-elevation myocardial infarction (STEMI), and chronic coronary syndrome (CCS).
The authors enrolled 167 patients (122 with STEMI, 45 with CCS) and showed a significant higher monocyte expression of PLIN2 in STEMI patients suggesting a potential involvement in plaque instability.
Furthermore, they found a significant higher proteasome activity in STEMI as compared to CCS patients, but significant inverse correlations were evident between PLIN2 levels and proteasome activity in the CCS groups.
The study research topic warrants careful consideration, due to strong interest about key pathophysiological mechanism in plaque vulnerability stratified for acute or chronic coronary scenarios.
Overall, this is a quite nicely written article with good statistic methodology.
However, it has some limitations that should be addressed from the authors:
- The major issue of the paper is related to the small sample size and the sample size imbalance with a substantially smaller CCS group (n=45) than the STEMI group (n=122), which may reduce statistical power for some subgroup analyses and increase the risk of type II errors.
- Along the previous point, the authors included only STEMI patients as acute scenario, but, as they know, a significant difference could emerge also in the subgroup analysis with unstable angina and NSTEMI. If possible, please include also these subtypes of ACS patients.
- The lipidomic analysis was performed in only 10 patients per group, limiting the generalizability and robustness of this specific result.
- While CD14+ monocytes were isolated, proteasome activity was evaluated on PBMCs rather than purified monocytes, introducing potential bias due to heterogeneous cell populations.
- Integration with intravascular imaging (OCT or IVUS) to correlate PLIN2 levels with plaque morphology or vulnerability could strengthen the translational value.
- In the paper it is not reported the prevalence of atrial fibrillation and consequently the proportion of patients treated with oral anticoagulant. This is a really relevant aspect because the presence of anticoagulant therapy, especially if not balanced between the two groups could impact the results. Furthermore, the authors should expand the discussion about the peri-procedural and post-procedural strategies (including imaging-guided PCI) to optimize outcomes in patients with atrial fibrillation undergoing PCI, even more in patients with higher plaque instability related to PLIN2 higher expression. In this respect, the authors should cite one recent relevant paper (doi:10.3390/jcdd12040142).
- The study provides indirect evidence for PLIN2 proteasomal escape but does not elucidate the specific post-translational modifications or molecular chaperones involved. Similarly, chaperone-mediated autophagy (CMA) is ruled out based on LAMP2A levels alone, without functional assays.
- Being observational and cross-sectional, causality cannot be established between PLIN2 expression and coronary instability or plaque events.
- Tracking PLIN2 levels over time and in response to treatment (statins, anti-inflammatory therapy) could clarify whether it is a modifiable risk marker.
- In the manuscript it’s not clear the clinical impact of these results. Do the authors suggest that all patients could benefit from a PLIN2 dosage to better clarify the ischemic risk? Could PLIN2 expression become a treatment modifier with the aim to intensify antithrombotic and lipid-lowering therapy in patients with high levels of PLIN2 expression? Please clarify and expand this aspect.
Author Response
In this prospective study, Dr. Francesco Canonico and colleagues aimed to investigate the role of Perilipin-2 (PLIN2) and its regulation mechanisms in atherosclerosis and plaque instability in patients with a diagnosis of ST-elevation myocardial infarction (STEMI), and chronic coronary syndrome (CCS).
The authors enrolled 167 patients (122 with STEMI, 45 with CCS) and showed a significant higher monocyte expression of PLIN2 in STEMI patients suggesting a potential involvement in plaque instability.
Furthermore, they found a significant higher proteasome activity in STEMI as compared to CCS patients, but significant inverse correlations were evident between PLIN2 levels and proteasome activity in the CCS groups.
The study research topic warrants careful consideration, due to strong interest about key pathophysiological mechanism in plaque vulnerability stratified for acute or chronic coronary scenarios.
Thank you for the kind comment.
Overall, this is a quite nicely written article with good statistic methodology.
Thank you for the kind comment.
However, it has some limitations that should be addressed from the authors:
The major issue of the paper is related to the small sample size and the sample size imbalance with a substantially smaller CCS group (n=45) than the STEMI group (n=122), which may reduce statistical power for some subgroup analyses and increase the risk of type II errors.
Along the previous point, the authors included only STEMI patients as acute scenario, but, as they know, a significant difference could also emerge in the subgroup analysis with unstable angina and NSTEMI. If possible, please include also these subtypes of ACS patients.
We thank the reviewer for this important observation. We fully acknowledge that the limited sample size and the imbalance between the CCS and STEMI groups represent a potential limitation of the study, which may affect the statistical power of some subgroup analyses.
Regarding the inclusion of NSTE-ACS patients, we recognize that their exclusion might also be perceived as a limitation. However, this decision was made intentionally, as the NSTE-ACS population is highly heterogeneous with respect to clinical presentation, underlying mechanisms of plaque instability, and time of presentation. Including such patients could have introduced additional variability and potential biases in the interpretation of our findings. By contrast, STEMI patients represent a more homogeneous group in terms of acute presentation and pathophysiology, thereby allowing for a clearer interpretation of the data.
We have now explicitly addressed both the sample size issue and the rationale for excluding NSTE-ACS patients in the revised Limitations paragraph (page 4, lines 193-198).
The lipidomic analysis was performed in only 10 patients per group, limiting the generalizability and robustness of this specific result.
We thank the reviewer for this important comment. We fully acknowledge that the lipidomic analysis was performed on a relatively small subset of patients (10 per group), which indeed represents a limitation and may restrict the generalizability and robustness of these specific findings. This choice was primarily driven by the complexity, cost, and resource requirements associated with detailed lipidomic profiling. Nevertheless, we believe that even with the limited sample size, the analysis provides valuable exploratory insights and supports the overall biological plausibility of our findings.
To ensure transparency, we have explicitly acknowledged this limitation in the revised manuscript (page 5, lines 198-203), highlighting that further studies in larger and independent cohorts will be necessary to validate and strengthen these observations.
While CD14+ monocytes were isolated, proteasome activity was evaluated on PBMCs rather than purified monocytes, introducing potential bias due to heterogeneous cell populations.
We thank the reviewer for raising this important point. We fully acknowledge that evaluating proteasome activity on total PBMCs, rather than on purified CD14⁺ monocytes, represents a limitation of our study. The use of PBMCs introduces cellular heterogeneity, which could potentially bias the results, as different leukocyte subsets (e.g., lymphocytes, NK cells, and dendritic cells) may display distinct proteasomal activity profiles.
This methodological choice was primarily driven by technical and logistical considerations, since proteasome activity assays require relatively large numbers of viable cells, which could not be consistently obtained from purified monocyte fractions in all patients. Importantly, previous studies have also employed PBMCs as a reliable surrogate model to evaluate proteasome function in the context of cardiovascular and inflammatory diseases [1. Mazzola A, Cianti R, Bini L, et al. Using peripheral blood mononuclear cells to determine proteome profiles in human cardiac failure. Eur J Heart Fail. 2008;10(8):749-757. doi:10.1016/j.ejheart.2008.06.003; 2. Georgiopoulos G, Makris N, Laina A, et al. Cardiovascular Toxicity of Proteasome Inhibitors: Underlying Mechanisms and Management Strategies: JACC: CardioOncology State-of-the-Art Review. JACC CardioOncol. 2023;5(1):1-21. Published 2023 Feb 21. doi:10.1016/j.jaccao.2022.12.005; 3. Kammerl IE, Hardy S, Flexeder C, et al. Activation of immune cell proteasomes in peripheral blood of smokers and COPD patients: implications for therapy. Eur Respir J. 2022;59(3):2101798. Published 2022 Mar 3. doi:10.1183/13993003.01798-2021; 4. Bramasole L, Meiners S. Profiling Proteasome Activities in Peripheral Blood – A Novel Biomarker Approach Journal of Cell Immunology, 2022], thus supporting the validity of our approach, while recognizing its limitations.
To ensure clarity and transparency, we have explicitly discussed this issue in the revised Limitations section (page 4, lines 186-189), and we now cite relevant literature to strengthen the rationale for our methodological choice.
Integration with intravascular imaging (OCT or IVUS) to correlate PLIN2 levels with plaque morphology or vulnerability could strengthen the translational value.
We sincerely thank the reviewer for this valuable suggestion. We fully agree that integrating intravascular imaging techniques such as OCT or IVUS would considerably enhance the translational relevance of our findings by allowing direct correlations between PLIN2 levels and plaque morphology or features of vulnerability. Unfortunately, intravascular imaging was not systematically performed in our patient cohort and could therefore not be included in the present analysis.
We acknowledge this as an important limitation and have now emphasized it explicitly in the revised Limitations section (page 5, lines 203-208). We also recognize this approach as a promising avenue for future studies, where the combination of molecular markers such as PLIN2 with high-resolution intravascular imaging could provide deeper mechanistic insights and potentially improve risk stratification in patients with coronary artery disease.
In the paper it is not reported the prevalence of atrial fibrillation and consequently the proportion of patients treated with oral anticoagulant. This is a really relevant aspect because the presence of anticoagulant therapy, especially if not balanced between the two groups could impact the results. Furthermore, the authors should expand the discussion about the peri-procedural and post-procedural strategies (including imaging-guided PCI) to optimize outcomes in patients with atrial fibrillation undergoing PCI, even more in patients with higher plaque instability related to PLIN2 higher expression. In this respect, the authors should cite one recent relevant paper (doi:10.3390/jcdd12040142).
We thank the reviewer for this thoughtful and important comment. As correctly pointed out, anticoagulant therapy could indeed influence outcomes and potentially impact the interpretation of our results. We would like to clarify that Table 1 of the manuscript reports the number and percentage of patients receiving oral anticoagulants among the pharmacological treatments. As shown in the table, the prevalence of anticoagulant therapy was balanced between STEMI and CCS patients, thus minimizing the risk of bias in this respect.
In line with the reviewer’s suggestion, we also agree that discussing peri- and post-procedural strategies in patients with atrial fibrillation undergoing PCI adds important clinical value. We have therefore expanded the Discussion section to include considerations on strategies such as imaging-guided PCI and optimized antithrombotic regimens in this specific subgroup, particularly in the context of patients with higher plaque instability potentially related to increased PLIN2 expression. Furthermore, we have incorporated the suggested recent reference (doi:10.3390/jcdd12040142) to strengthen this discussion.
The study provides indirect evidence for PLIN2 proteasomal escape but does not elucidate the specific post-translational modifications or molecular chaperones involved. Similarly, chaperone-mediated autophagy (CMA) is ruled out based on LAMP2A levels alone, without functional assays.
We thank the reviewer for this insightful comment. We agree that our study provides indirect evidence of PLIN2 proteasomal evasion without identifying the specific post-translational modifications or molecular chaperones involved. Similarly, while we assessed LAMP2A expression, we acknowledge that this does not substitute for functional CMA assays.
As noted in the “Study limitations,” we did not perform analyses of post-transcriptional or post-translational regulation, nor functional CMA studies. Our findings should therefore be considered hypothesis-generating. Future work will focus on defining the molecular determinants of PLIN2 degradation, including the role of post-translational modifications, molecular chaperones, and CMA activity.
Being observational and cross-sectional, causality cannot be established between PLIN2 expression and coronary instability or plaque events.
Tracking PLIN2 levels over time and in response to treatment (statins, anti-inflammatory therapy) could clarify whether it is a modifiable risk marker.
We thank the reviewer for this valuable observation. We fully acknowledge that, given the observational and cross-sectional nature of our study, causality cannot be established between PLIN2 expression and coronary instability or plaque events. The suggestion of longitudinal assessment of PLIN2 levels in relation to treatment (e.g., statins, anti-inflammatory therapy) is indeed of great interest, though it was beyond the scope of the present work. We agree that such an approach could clarify whether PLIN2 represents a modifiable risk marker and could serve in risk stratification. This point has now been incorporated into the conclusion (page 5, lines 226-227) as a potential clinical implication of our findings and into the limitations section (page 5, lines 208-212).
In the manuscript it’s not clear the clinical impact of these results. Do the authors suggest that all patients could benefit from a PLIN2 dosage to better clarify the ischemic risk? Could PLIN2 expression become a treatment modifier with the aim to intensify antithrombotic and lipid-lowering therapy in patients with high levels of PLIN2 expression? Please clarify and expand this aspect.
Thank you for your comment. As previously pointed out, “The potential clinical implications of PLIN2 regulation may include its potential prognostic and therapeutic use, for example as an early identification and risk stratification for the onset and progression of coronary artery disease or as a potential biomarker for better management of patient drug treatments.”
In particular, the modulation of PLIN2 expression could help in distinguishing patients at higher risk of developing coronary events at an earlier stage, thus allowing more tailored preventive strategies. Moreover, by serving as a biomarker linked to lipid metabolism and vascular pathology, PLIN2 may guide the optimization of pharmacological therapies, improving patient stratification and potentially contributing to personalized treatment approaches.
Round 2
Reviewer 2 Report
Comments and Suggestions for Authors
This is a second revision of well executed research and a well written paper aimed to investigate the role of Perilipin-2 (PLIN2) and its regulation mechanisms in atherosclerosis and plaque instability in patients with a diagnosis of ST-elevation myocardial infarction (STEMI), and chronic coronary syndrome (CCS).
The paper is well revised and the authors provided satisfactory comments to my suggestions.
There are no concerns about ethical issues, conflict of interest and plagiarism/publication ethics.